# The Adeno-Associated Virus Replication Protein Rep78 Contains a Strictly C-Terminal Sequence Motif Conserved Across Dependoparvoviruses

**DOI:** 10.3390/v16111760

**Published:** 2024-11-12

**Authors:** David G. Karlin

**Affiliations:** 1Division Phytomedicine, Thaer-Institute of Agricultural and Horticultural Sciences, Humboldt-Universität zu Berlin, Lentzeallee 55/57, D-14195 Berlin, Germany; davidgkarlin@gmail.com; 2Independent Researcher, 13000 Marseille, France

**Keywords:** Adeno-Associated Virus, C-terminal motif, C-terminome, minimotif, mini-motif, eukaryotic linear motif, Alphafold3, NS1

## Abstract

Adeno-Associated Viruses (AAVs, genus *Dependoparvovirus*) are the leading gene therapy vector. Until recently, efforts to enhance their capacity for gene delivery had focused on their capsids. However, efforts are increasingly shifting towards improving the viral replication protein, Rep78. We discovered that Rep78 and its shorter isoform Rep52 contain a strictly C-terminal sequence motif, DDx3EQ, conserved in most dependoparvoviruses. The motif is highly negatively charged and devoid of prolines. Its wide conservation suggests that it is required for the life cycle of dependoparvoviruses. Despite its short length, the motif’s strictly C-terminal position has the potential to endow it with a high recognition specificity. A candidate target of the DDx3EQ motif might be the DNA-binding interface of the origin-binding domain of Rep78, which is highly positively charged. Published studies suggest that this motif is not required for recombinant AAV production, but that substitutions within it might improve production.

## 1. Introduction

Adeno-Associated Viruses (AAVs, genus *Dependoparvovirus*) are the leading vector for delivering gene therapies [1,2,3,4]. Recombinant AAVs can package foreign genes into their capsid [1,5,6], and, until recently, efforts to enhance gene delivery had focused on tailoring and improving capsids. However, efforts are increasingly shifting to improving the viral replication protein encoded by the rep gene [7,8,9].

Rep encodes four protein isoforms (Figure 1) thanks to a combination of alternative promoters and alternative splicing sites [10]: two long isoforms (Rep78 and Rep68) and two short isoforms (Rep52 and Rep40). The larger Rep proteins, Rep78 and Rep68, are required to replicate the genome, while Rep52 and Rep40 facilitate the packaging of the genome [11]. Rep78 and Rep68 are sufficient for recombinant AAV production [12].

Three main regions have been delineated in Rep78 (Figure 1): an origin-binding domain [13], a helicase domain [14], and a C-terminal region, predicted to contain zinc-fingers [15]. While the origin-binding and helicase domains have been systematically investigated, there has been no in-depth sequence analysis of the C-terminus beyond the putative identification of three zinc fingers [15]. Here, we examined sequences of Rep 78 across all dependoparvoviruses, beyond the usual ones employed in gene therapy (AAV1 to AAV13), and discovered that Rep78 contains a strictly C-terminal motif conserved in most dependoparvoviruses.

The representation is to scale. Znf: Zinc finger. The DDx3EQ motif was discovered in the present study. The numbering of all Rep proteins is given with reference to Rep78.

## 2. Materials and Methods

### 2.1. Protein Sequence Analysis

We extracted *Dependoparvovirus* sequences from NCBI’s Genbank [16] on 1 July 2024. We used Psi-Coffee [17] for multiple sequence alignment. Alignments are shown with Jalview [18] using the ClustalX colouring scheme [19]. We used flDPnn [20] for predicting disordered regions.

The compositional bias of the DDx3EQ motif was assessed using Composition Profiler [21] against two datasets, (1) SwissProt (version 51) [22] and (2) a dataset composed of all dependoparvovirus Rep78 sequences (available in Appendix A), after having removed their DDx3EQ motif, i.e., the last C-terminal 7 aa in each sequence.

### 2.2. Sequence Motif Searches

We looked for known motifs similar to the DDx3EQ motif using Comparimotif [23] and TOMtOM [24].

We used Comparimotif to scan the databases ELM [25] (March 2022 release) with the regular expression [DEP]D[^P][^P][^P]EQ$, in which [^P] corresponds to any aa except P, and ‘$’ specifies that the motif must be a C-terminal. The request was made through a restful API: https://slim.icr.ac.uk/restapi/rest/get/comparimotif?task=run_comparimotif&motif=[DEP]D[^P][^P][^P]EQ$, accessed on 19 August 2024.

We also used TOMtOM [24] to scan the database Prosite (april 2021 release) [26].

We looked for proteins containing the DDx3EQ motif using Patternsearch, ran from the web-based version of the MPI toolkit [27] (https://toolkit.tuebingen.mpg.de/), accessed on 5 August 2024, against three databases that are subsets from Genbank [16]: (1) the database nr_vir70_12Mar containing viral proteins clustered at 70% sequence identity on 12 March 2024; (2) the database Homo Sapiens_4Jul containing *Homo Sapiens* proteins on 4 July 2024; (3) and the database PDB_nr_12_Mar containing proteins with an experimentally solved 3D structure on 12 March 2024. We used, as input, the regular expression [DEP]-D-{P}-{P}-{P}-E-Q>, which follows the Prosite syntax [28], in which {P} corresponds to an excluded P aa and ‘>’ specifies that the motif must be C-terminal.

### 2.3. D Structure Prediction and Visualization

We predicted the 3D structure of the C-terminal region of AAV2 Rep78 using Alphafold3 [29] with 3 zinc atoms. AlphaFold3 outputs a measure of reliability of the 3D structure for each aa, pLDDT. pLDDT ≥ 0.70 corresponds to a reliable prediction and pLDDT ≥ 0.90 corresponds to a highly reliable prediction (expected to be competitive with an experimentally solved 3D structure) [30]. Structures were visualized using ChimeraX (version 1.8) [31].

We also used Alphafold3 to predict the 3D structure of a putative complex between the origin-binding domain and the C-terminal DDx3EQ peptide of AAV2 Rep78 (D^615^DCIFEQ^621^). Alphafold3 provides a measure of reliability of the interaction, ipTM. An ipTM > 0.8 indicates a reliably predicted interaction, 0.8 ≥ ipTM ≥ 0.6 corresponds to a “gray zone” in which predictions may be correct or incorrect, and ipTM < 0.6 indicates an unreliable prediction [32,33].

## 3. Results

### 3.1. The C-Terminal Region of Rep78 Contains 3 Predicted Zinc Fingers and Flexible Regions

We analyzed the Rep78 protein of AAV2, the *Dependoparvovirus* model species (Genbank accession number YP_680423.1, see Table 1). The C-terminal region of Rep78 starts with a linker predicted to be disordered (aa 493–521 in AAV2, see Figure 1). We modelled the 3D structure of the remaining C-terminal part (aa 522–621) using Alphafold3 [29] (the coordinates of the model are in Appendix A). The model contains two regions reliably predicted to adopt a fixed 3D structure (in red in Figure 2A; see also Figure 1):(1)aa 525–573 are composed of two zinc fingers (named 1 and 2) of the CHCC type (Figure 2A, left). These zinc fingers are predicted to adopt a fixed conformation relative to each other (Figure 2B).(2)aa 587–612 are composed of a third zinc finger, also of the CHCC type, followed by a predicted α-helix (Figure 2A, right).

All of the three zinc fingers follow the consensus sequence C-x(2)-H-x(n)-C-x(2)-C, corresponding to an unusual type of zinc finger, found, for example, in the mengovirus Leader protein and in archaeal transcription factors [34], and divergent from classical zinc fingers [35,36].

The remaining regions are not reliably modelled using Alphafold3, despite being predicted to be ordered, which indicates that they are conformationally flexible; they are visible as blue or white ribbons in Figure 2.

### 3.2. The C-Terminal Region of Dependoparvoviral Rep78 and Rep52 Contains a Conserved Motif, DDx3EQ, Not Similar to a Known Motif

The C-terminal region of Rep78 is highly variable in sequence across dependoparvoviruses, as shown in Figure 3 (see also Appendix A). However, we noticed that in almost all dependoparvoviruses, it contains a D-D-x(3)-E-Q sequence motif (in which x(3) represents a consecutive stretch of any three aa) at the C-terminus (aa 615–621 in AAV2). The motif is shown in Figure 3, right, and for simplicity, we will refer to it as DDx3EQ.

In all dependoparvoviruses, the last aa of the motif, Q, is also the last aa of Rep78. This strictly C-terminal position confers a markedly enhanced specificity to motifs [37] (see Section 4).

Only a handful of dependoparvovirus Rep78 proteins do not have the DDx3EQ motif (Appendix A), being the related viruses desmodus rotundus dependoparvovirus (*Dependoparvovirus chiropteran2*) [38] and feline dependoparvovirus (*Dependoparvovirus carnivoran1*) [39], the canary dependoparvoviruses 1 and 2 [40], and five bird dependoparvoviruses [41]: isolates ltt164par2 (Genbank accession number QLF86430.1), sis142par1 (QKE54964.1), zftwig05par3 (QKN88780.1), wpk049par01 (QKE60686.1), and avian AAV isolate BR_DF12 (YP_010802670.1). The latter presents a striking case. Its rep gene contains a long (1803 nucleotides) reading frame overlapping that of Rep78, which encodes a potential protein of 243 aa ending with a C-terminal DDx3EQ motif. The sequence of that protein and its location within Rep are presented in Appendix A.

Finally, we found that the DDx3EQ motif is not similar to a known motif, according to both Comparimotif [23] and TOMtOM [24] (see Section 2).

**Figure 3 viruses-16-01760-f003:**
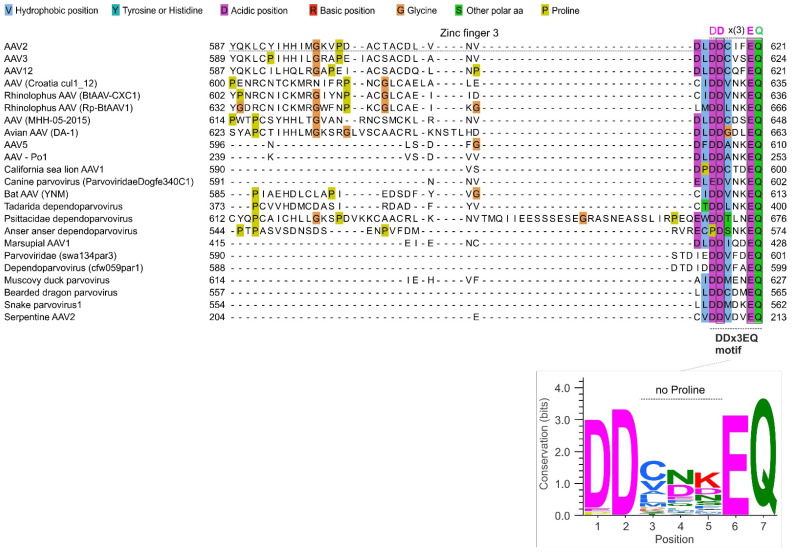
The variable C-terminal region of dependoparvovirus Rep78 contains a DDx3EQ motif. Top panel: Sequence alignment of the C-terminal region of Rep78 among representative dependoparvoviruses. Note its high variability and the conserved DDx3EQ motif at the very C-terminus. Bottom panel: sequence logo of the DDx3EQ motif, made using WebLogo [42].

### 3.3. The Motif Contains Three Strictly Conserved aa, Is Highly Negatively Charged, and Is Devoid of Prolines

The frequency of each aa at each position of the motif is shown in Figure 3, bottom panel. Three positions are strictly conserved (Figure 3, bottom panel): an aspartate in position 2 (D616 in AAV2), a glutamate in position 6 (E620), and a glutamine in position 7 (Q621). Position 1 almost exclusively contains an aspartate (D615), rarely a glutamate (also negatively charged) or a proline. Position 3 is enriched in hydrophobic aa; position 4 is enriched in negatively charged aa, depleted in hydrophobic aa, and contains no positively charged aa; and position 5 is enriched in polar aa, particularly in charged ones.

The motif is significantly (*p* < 0.005) enriched in negative aa compared both to the protein database SwissProt and to the rest of Rep78 (see Section 2); its negative charge is expected to be further increased by its C-terminal carboxylate ion (COO^−^).

Strikingly, the motif is completely devoid of prolines, except at position 1 in anser anser dependoparvovirus (Figure 3) and a few related species, suggesting that forming an α-helix might be required for its function.

### 3.4. A Conserved DDx3EQ Motif Is Found in One Protein from a Eukaryotic Virus and in One Human Protein

To obtain clues regarding the function of the DDx3EQ motif, we searched for other proteins from either eukaryotic viruses or humans that would have the motif conserved in at least another species (see Section 2).

In eukaryotic viruses other than dependoparvoviruses, we could only identify one protein with a conserved C-terminal DDx3EQ motif, the protease 2A from the genus *Enterovirus*. As an example, the C-terminus of the protease 2A of enterovirus D (NP_740416.1) is EDdamEQ, i.e., with an E in position 1 instead of the D most commonly found in dependoparvovirus Rep78—see Figure 3, bottom panel).

The motif is found in the species *Enterovirus A–B*, *D*, *F*, and *H–J*, but not in *Enterovirus E*, G, and *K*, nor in the three species *Rhinovirus A*, *B*, and *C*. In *Enterovirus C*, the motif is degenerate, i.e., there is an E in position 2 instead of the strictly conserved D. Appendix A presents an alignment of enterovirus proteases 2A that have the DDx3EQ motif.

The enteroviral protease 2A is cleaved immediately after the conserved Q of the motif by another enteroviral protease, 3C [43]. Apart from this Q, no position of the DDx3EQ motif corresponds to the cleavage specificity of 3C, whose main specificity determinant is an A three aa upstream of the Q at which the cleavage occurs (i.e., in position 4 of the motif) [44]. Therefore, the presence of the motif in the 2A protease does not stem from a requirement for cleavage by the 3C protease. Interestingly, removing the 5 aa immediately upstream of the C-terminal Q (i.e., most of the motif) from the 2A protease of poliovirus (*Enterovirus C*) is lethal without affecting its protease function [45].

The DDx3EQ motif forms a coil with no regular secondary structure in the 2A protease of coxsackievirus B4 (*Enterovirus B*) [46], similar to the Alphafold3 prediction for the AAV2 Rep78 motif (Figure 2A). The motif is not visible in the structure of the related coxsackievirus B3 2A protease, suggesting that it is flexible [47].

Finally, we could only identify a single human protein with a conserved C-terminal DDx3EQ motif: Cep57L1 (Centrosomal protein 57 kDa-like protein 1, Uniprot accession number Q8IYX8). Its last 7 C-terminal aa are DDimwEQ. The motif is conserved across amniotes (clade *Amniota*). Appendix A presents an alignment of Cep57L1 orthologs that have the DDx3EQ motif. Cep57L1 contributes to maintaining centriole engagement during interphase [48]. No functional data are available regarding the role of its C-terminus, to our knowledge, and in a recent study, Cep57L1 was not part of the proteins identified as having mutations in their C-terminus that cause disease in humans [49].

## 4. Discussion

### 4.1. The DDx3EQ Motif Should Have a High Binding Specificity Despite Its Short Length, and Is Probably Essential for AAV Replication

Only a handful of strictly C-terminal sequence motifs have been described in eukaryotic viruses [37,50]. The C-terminal position confers a high binding specificity to these motifs, even when relatively short, because only one free carboxy group is found in each protein at the C-terminus, where it can be recognized by specialized enzymes. For example, with the average length of a human protein being ~600aa, a motif containing a glutamine with a free C-terminal carboxy group is found 600 times less frequently than a glutamine *within* a non C-terminal motif [37].

Given the high rate of evolution of viral proteins, the DDx3EQ motif is most probably essential for dependoparvoviruses, since it is conserved in almost the whole genus. We cannot infer its function from published experimental studies, since, to our knowledge, no study tested the effect of substitutions or deletions of the very C-terminus of Rep78 (aa 608–621, beyond zinc finger 3) on the replication of wild-type AAVs. The further downstream substitution we are aware of is in aa 607 in zinc finger 3 [15], and the second most downstream substitution we are aware of concerns aa 540 [51].

Interestingly, we could only identify a single viral protein (the *enterovirus* protease 2A) and a single human protein (Cep57L1) with a conserved DDx3EQ motif. We note that the presence of the motif in these proteins may be coincidental and does not imply functional similarity to Rep78.

### 4.2. Hypothesis: The DDx3EQ Motif May Bind the DNA-Binding Interface of the Origin-Binding Domain of Rep78

“No one believes a hypothesis except its originator, but everyone believes an experiment except the experimenter” (William Ian Beardmore Beveridge). A study on human C-terminal motifs found that they typically have either one of three functions, in decreasing order of frequency [52]: (1) directing post-translational modification [53]; (2) binding another protein(s); (3) directing trafficking through the cell [37]. We can only provide a meaningful hypothesis regarding the second function, i.e., binding a protein.

Given the considerable rate of sequence evolution of viral proteins, the fact that such a short motif contains three strictly conserved aa suggests that it binds either a cellular protein or a highly conserved region of a viral protein. A prime candidate would be the DNA-binding interface of the origin-binding domain of Rep78, which is well conserved in sequence and positively charged, while the DDx3EQ motif is highly negatively charged. This hypothesis is biologically meaningful since (a) the DDx3EQ motif and the origin-binding domain are always in close proximity, being part of the same protein, and (b) binding the motif would provide a mechanism for regulating the interaction of this domain with the inverted terminal repeats during the replication cycle [54]. In this scenario, the DDx3EQ motif of Rep 52, the shorter isoform of Rep78, could not interact in *cis* with the origin-binding domain, since Rep52 is devoid of this domain (Figure 1). We emphasize that this scenario is merely proposed as a biological hypothesis meant to guide experiments.

Attempts to confirm or infirm our hypothesis using Alphafold3 were unsuccessful; Alphafold3 output a reliability estimator (ipTM) of only 0.55 for the interaction between the origin-binding domain and the DDx3EQ peptide of AAV2 Rep78. This value indicates an unreliable prediction (see Section 2) which cannot determine whether the complex exists or not.

### 4.3. The DDx3EQ Motif Might Not Be Necessary for Recombinant AAV Production, but Substitutions Within It Improve Production

The DDx3EQ motif is found in most taxa of AAVs relevant for gene therapy [55], i.e., AAV 1–4 and 6–13 (*Dependoparvovirus primate 1*), AAV5 (*Dependoparvovirus mammalian1*), and porcine AAV1 (unclassified) [56]. A recent study found that in the absence of Rep78, Rep68 was not sufficient for efficient recombinant AAV production [8], indicating that the C-terminal region of Rep78 is also required. It would be interesting to determine whether the DDx3EQ motif in particular contributes to this requirement.

In that regard, a recent study systematically tested the effect of all single aa substitutions in Rep78 and Rep68 on the production of recombinant AAVs [9]. Substitutions of conserved positions of the motif or introduction of prolines (normally absent from the motif) did not result in significantly lower production, indicating that the DDx3EQ motif is not necessary for production of recombinant AAVs, at least in the conditions tested. Intriguingly, in that study, not only were some substitutions neutral, but most even had a mildly beneficial effect on recombinant AAV production (i.e., in Figure 2 of [9], the last 7aa of Rep78 form a red “patch”). We will detail these briefly below. This study tested two production platforms. In the first one, pCMV-Rep78/68, Rep68 and Rep78 were produced and mutated from one plasmid, and the other AAV proteins (Rep40, Rep52, and the capsid proteins) were produced from other plasmids. In the second platform, wtAAV2, all of the AAV proteins were produced from a single plasmid, and all four Rep proteins were thus mutated simultaneously.

Although numerous substitutions were mildly beneficial, only a few were *significantly* beneficial. In the first platform, no substitution had a significant fitness effect. In the wtAAV2 platform, three substitutions significantly improved production: I618T, affecting position 4 (T being observed at this position in some dependoparvoviruses, see Figure 2, bottom panel); F619N, affecting position 5 (N being seen at this position in some dependoparvoviruses); and E620S, which affects the strictly conserved E in position 6.

In summary, the DDx3EQ motif might not be necessary for efficient production of recombinant AAVs, but substitutions within it have the potential to improve production. Note that *recombinant* AAV production, as measured in [9], and *wild-type* AAV replication are not identical processes. As such, it is possible that the DDx3EQ motif may be essential for wild-type AAV replication, but dispensable for recombinant AAV production.

### 4.4. Sequence Motifs Can Be Identified Even Within Highly Variable Protein Regions by Examining Alignment of Orthologs

As Figure 2 makes clear, the DDx3EQ motif is clearly visible in the alignment of *Dependoparvovirus* Rep78, even by a non-expert. Many such motifs can be identified in viral proteins using simple visual examination (e.g., soyuz1 and soyuz2 in *Paramyxovirinae* [57]).

Conversely, these motifs are not detectable even with advanced homology detection software commonly used to ascribe functions to viral proteins [58] (such as PSI-BLAST [59] or HHpred [60]), because they are too short (7–20 aa) and “hidden” within a highly variable region. Therefore, we recommend systematically aligning variable regions of orthologous proteins across suitable evolutionary distances (i.e., genus or subfamily) and examining them for conserved sequence motifs.

## Figures and Tables

**Figure 1 viruses-16-01760-f001:**
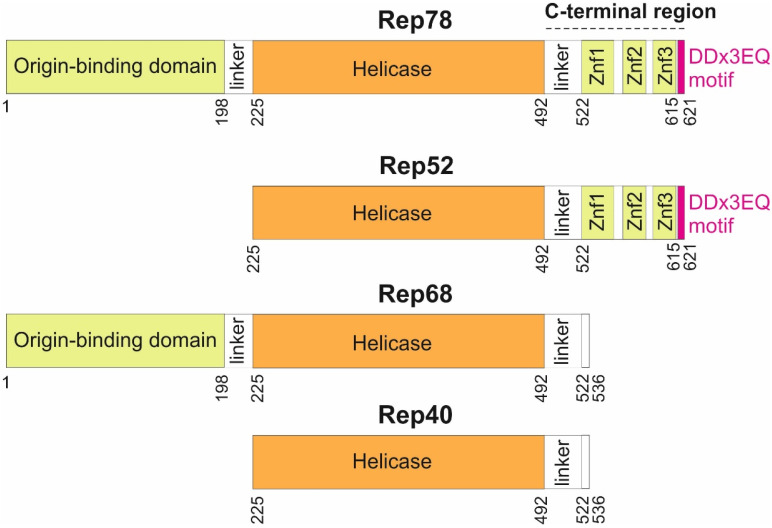
Domain organization of the four proteins produced from the AAV2 rep gene.

**Figure 2 viruses-16-01760-f002:**
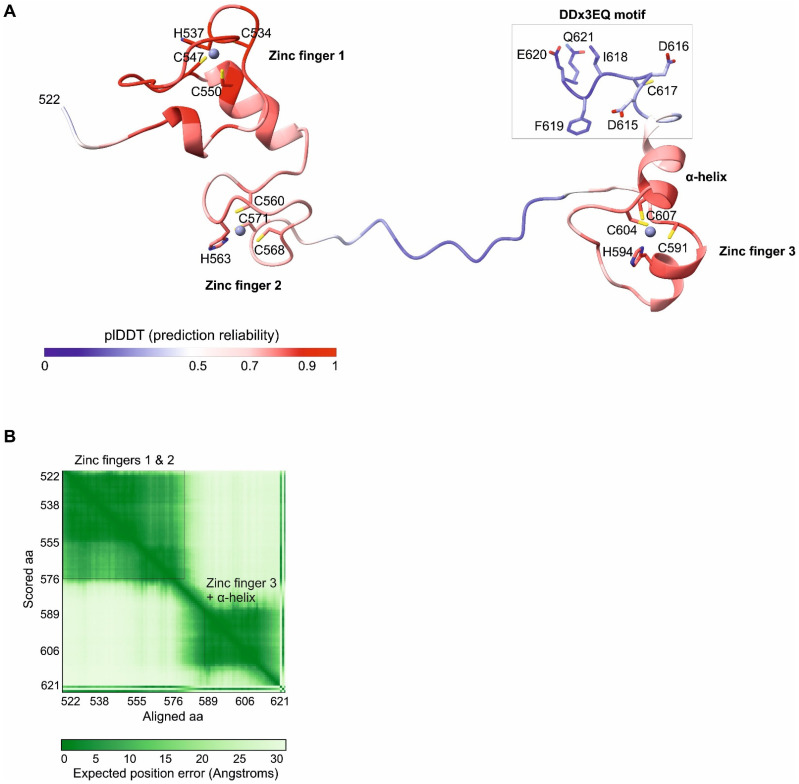
Predicted 3D structure of the C-terminal region of AAV2 Rep78 (aa 522–621). (**A**) Structure predicted using Alphafold3. Zinc ions are pictured as spheres. Regions for which Alphafold3 did not predict a fixed structure are represented as blue or white ribbons. In particular, this is the case of the C-terminal DDx3EQ motif (see text), whose aa are only shown as an illustration, since their position is not reliably predicted. (**B**) PAE (Predicted Alignment Error). Green rectangles represent the regions of Rep78 in which all aa are predicted to have a fixed conformation with respect to each other.

**Table 1 viruses-16-01760-t001:** Rep78 proteins presented in Figure 2.

Common Name	Species or Taxon	Genbank Accession Number
AAV2	*Dependoparvovirus primate1*	YP_680423.1
AAV3	*Dependoparvovirus primate1*	NP_043940
AAV5	*Dependoparvovirus mammalian1*	YP_068408.1
AAV12	*Dependoparvovirus primate1*	DQ813647
AAV (isolate Croatia cul1_12)	Unclassified	QHY93489
AAV (isolate MHH-05-2015)	Unclassified	YP_009552823.1
AAV—Po1 [porcine AAV1]	Unclassified	ACN42943.1
Anser anser dependoparvovirus	Unclassified	QTE04020.1
Avian AAV (strain DA-1)	*Dependoparvovirus avian1*	YP_077182.1
Bat AAV (strain YNM)	*Dependoparvovirus chiropteran1*	YP_003858571.1
Bearded dragon parvovirus	*Dependoparvovirus squamate2*	YP_009154712.1
California sea lion AAV1	*Dependoparvovirus pinniped1*	YP_009507366.1
Canine parvovirus (isolate ParvoviridaeDogfe340C1)	Unclassified ^(1)^	WDW25820.1
Dependoparvovirus (isolate cfw059par1)	Unclassified	QKN88755.1
Marsupial AAV1	Unclassified	AZP54391.1
Muscovy duck parvovirus	*Dependoparvovirus anseriform1*	YP_068410.1
Parvoviridae (isolate swa134par3)	Unclassified	QKE54950.1
Psittacidae dependoparvovirus	Unclassified	QTE03943.1
Rhinolophus pusillus AAV (isolate BtAAV-CXC1)	Unclassified	QDX47269.1
Rhinolophus pusillus AAV1 (isolate Rp-BtAAV1_34C_MJ_YN_2012)	Unclassified	ATV81500.1
Serpentine AAV2	Unclassified	ACJ66590.1
Snake parvovirus 1	*Dependoparvovirus squamate1*	YP_068093.1
Tadarida brasiliensis associated dependoparvovirus	Unclassified	UJO02142.1

AAV: Adeno-associated virus. (1) Erroneously classified as *Protoparvovirus carnivoran1* in Genbank.

## Data Availability

Data are contained within the article and Appendix A.

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
