# Peer review of "The Adeno-Associated Virus Replication Protein Rep78 Contains a Strictly C-Terminal Sequence Motif Conserved Across Dependoparvoviruses"

_viruses, 2024, doi:10.3390/v16111760_

Round 1

Reviewer 1 Report

Comments and Suggestions for Authors

The manuscript by Karlin focuses on the analysis of the Adeno-Associated Virus (AAV) Rep protein sequence, identifying a novel motif at the C-terminal end of Rep78 and Rep52, named DDx3EQ, which is conserved across all dependoparvoviruses. The first part of the paper is a review of the Alphafold3 structure prediction for Rep78, specifically examining the C-terminal region from residues 522 to 621. This prediction confirmed the presence of three putative zinc finger motifs of the type CHCC. Subsequently, the author performed a protein sequence analysis of dependoparvoviruses and discovered a conserved motif consisting of 7 residues, identified as DDx3EQ. According to Comparimotif and TOMtOM, this motif is novel. Next, a search in protein databases found that this motif is present in the protease 2A of enterovirus D and in the human centrosomal protein 57 kDa-like 1, however, no function has been attributed to this region.

   Although this manuscript does not provide experimental data to determine the role of this motif in the AAV life cycle, a recently published study systematically tested the effect of all single amino acid substitutions in Rep proteins. The study showed that some mutations within this motif improve AAV production.

The manuscript presents new information about the structure of the C-terminal domain and identifies a novel sequence motif whose function is currently unknown. This work will serve as a foundation for future experiments. Although the AAV virus genome sequence has been known for over 50 years, this manuscript offers insights that could enhance our understanding of the function and regulation of AAV Rep proteins. There are several issues that will need to be addressed :

-              The author provides one hypothesis about the potential function of this motif to regulate the activity of the origin binding domain (OBD). Given that the author has used Alphafold3 for the prediction of Rep78 structure, the author could use it to also try to determine an interaction between the OBD and the DDx3EQ motif.

-              Page 1, line 29: The Rep protein is not a replicase in the strict sense.

-              Page 3, line 90: Figure 2A does not illustrate the interaction between the two zinc fingers.

-              Section 4.1 of the discussion is not clear about the meaning of high specificity.

Reviewer 2 Report

Comments and Suggestions for Authors

In the manuscript by Karlin et al, the author provides interesting findings that the adeno-associated virus replicase Rep78 may contains a strictly C-terminal sequence motif DDx3EQ that are conserved across dependoparvoviruses. The shown motif is highly negatively charged and maybe functions as a DNA binding motif. Overall, this manuscript provides novel information about a potential motif in dependoparvoviruses that may have biological functions during virus life cycle.

i) Minor points

1) lines 163-170, it is suggested to move this paragraph to discussion parts.

2) lines 206to 207, it is suggested to combine with lines 208 to 212.

3) lines 234 to 238, it is suggested to combine with lines 239 to 246.

Reviewer 3 Report

Comments and Suggestions for Authors

Karlin’s manuscript provides a detail analysis of the C-terminus of the Rep78 and Rep58 proteins of AAV2, a region of ~100 animo acids, identifying a conserved DDx3EQ motif at the end of the protein. Further analyses concluded this minimotif is strictly conserved across most Dependoparvovirus members. The author hypothesizes that the DDx3EQ motif may interact with the DNA-binding interface of Rep78's origin-binding domain, and suggests that while this motif appears not necessary for rAAV production, specific substitutions within it may enhance production efficiency. Given that rAAV is a promising gene therapy vehicle, understanding the structure and function of its replicase (Rep proteins) offers valuable insights for optimizing vector production—particularly as current research focuses primarily on modifying the capsid to enhance tropism. This manuscript will be of significant interest to the AAV field, and Reviewer recommends its publication in Viruses. This manuscript is well written, with only two minor suggestions

1)        Figure 1: It is better to spefify that the analysis was conducted on rAAV2 Rep proteins. Additionally, the animo acid numbering for Rep52/40 should begin at 1 rather than at 225, as numbered for Rep78/68, unless author points out this.

2)        Line 89 and 92. It will be helpful to either provide a reference for the CHCC-type zinc finger domain, or give a short description of this in the introduction section when taking the C-ter of Rep containing this type of zinc fingers
